# CORRELATIONS ARE RUINING YOUR GRADIENT DESCENT

## ABSTRACT

Herein the topics of (natural) gradient descent, data decorrelation, and approximate methods for backpropagation are brought into a common discussion. Natural gradient descent illuminates how gradient vectors, pointing at directions of steepest descent, can be improved by considering the local curvature of loss landscapes. We extend this perspective and show that to fully solve the problem illuminated by natural gradients in neural networks, one must recognise that correlations in the data at any linear transformation, including node responses at every layer of a neural network, cause a non-orthonormal relationship between the model's parameters. To solve this requires a method for decorrelating inputs at each individual layer of a neural network. We describe a range of methods which have been proposed for decorrelation and whitening of node output, and expand on these to provide a novel method specifically useful for distributed computing and computational neuroscience. Implementing decorrelation within multi-layer neural networks, we can show that not only is training via backpropagation sped up significantly but also existing approximations of backpropagation, which have failed catastrophically in the past, benefit significantly in their accuracy and convergence speed. This has the potential to provide a route forward for approximate gradient descent methods which have previously been discarded, training approaches for analogue and neuromorphic hardware, and potentially insights as to the efficacy and utility of decorrelation processes in the brain.

## 1 INTRODUCTION

The method of gradient descent is as popular as it is intuitive. This method, of stepping in the direction of steepest descent of a function, is applied successfully across the engineering sciences as well as in the modern AI revolution to find (local) optima of arbitrary functions. Alternative optimization methods have proven largely unsuccessful in being generally applied to continuous functions of arbitrary form, with second-order methods being largely brittle when applied to non-convex functions. Nonetheless, methods for speeding-up optimizations are not only interesting but have huge potential economic and environmental impact.

In 1998 it was proposed that there might be a perspective beyond typical gradient descent, called natural gradient descent (Amari, 1998), which might overcome some elements of skew and scale in the updates produced by gradient descent. Natural gradient descent has since been explored at the edges of the field of optimization, and deep neural network training (Bernacchia et al., 2018; Martens & Grosse, 2015; Desjardins et al., 2015; Heskes, 2000), with sometimes greater stability than traditional second order methods (Dauphin et al., 2014), though recently developed second order methods show significant promise (Gupta et al., 2018; Ren & Goldfarb, 2021; Vyas et al., 2024). Regardless, the principles of natural gradients are less widely understood, less applied, and less intuitive than they could be.

Simultaneous to this line of development, the principles behind learning in natural biological systems and potential algorithms for learning in distributed systems have been under investigation (Lillicrap et al., 2020). From these fields have sprung a whole range of approximate methods for gradient descent which promise to explain how learning might occur in brains or how it might be enabled in analogue hardware (neuromorphic, analogue, or otherwise).

We aim to bring together these lines of research, contributing to each individually while also providing a common space for impact. Specifically,

1. We demonstrate that correlations in data (between features) at the input and hidden layers of deep networks are one half of natural gradients, and that they contribute to a non-orthonormal basis in parameters,

2. We explore and expand the efficacy of methods for removing correlations from deep neural networks to better align gradient descent with natural gradients, and

3. We show that decorrelating mechanisms not only speed up learning by backpropagation, but that decorrelation can also enable alternatives approximations to gradient descent.

This paper is organised in an unconventional format due to the multiple sub-fields within which it is embedded and to which it contributes. Therefore, one should consider each of the coming sections as descriptions of a particular contribution, insight, or result embedded within a larger narrative. The titles of each section represent their core contribution to this narrative.

The intention of this work is to bring insight to those who wish for more efficient gradient descent and excitement to those interested in approximate gradient descent methods whether for explanation of learning in biological systems or implementation in physical/neuromorphic systems.

## 2 DATA CORRELATIONS CAN CAUSE PARAMETERS TO ENTER A NON-ORTHONORMAL RELATION

Here we address the first of our goals. We describe gradient descent, its relation to natural gradients, and demonstrate the often ignored aspect of input correlations impacting parameter orthonormality.

### 2.1 GRADIENT DESCENT

Consider the case in which we have a dataset which provides input and output pairs $(\boldsymbol{x}, \boldsymbol{y}) \in \mathcal{D}$, and we wish for some parameterised function, $\boldsymbol{z} = \boldsymbol{f_\theta}(\boldsymbol{x})$, with parameters $\boldsymbol{\theta}$ to produce a mapping relating these. This mapping would be optimal if it minimised a loss function $\mathcal{L}(\boldsymbol{\theta}) = \frac{1}{|\mathcal{D}|} \sum_{(\boldsymbol{x}, \boldsymbol{y}) \in \mathcal{D}} \ell(\boldsymbol{f_\theta}(\boldsymbol{x}), \boldsymbol{y})$, where the sample-wise loss function, $\ell$, can be a squared-error loss for regression, the negative log-likelihood for classification, or any other desired cost.

The problem which we wish to solve in general is to minimise our loss function and find the optimal set of parameters, effectively to find $\operatorname{argmin}_{\boldsymbol{\theta}} \mathcal{L}(\boldsymbol{\theta})$. However, finding this minimum directly is challenging for most interesting problems. Gradient descent proposes a first-order optimization process by which we identify an update direction (not directly the optimized value) for our parameters based upon a linearization of our loss function. This is often formulated as taking a 'small step' in the direction steepest descent of a function, in its gradient direction, such that

$$\delta\boldsymbol{\theta}^{\mathrm{GD}} = -\eta \nabla \mathcal{L}(\boldsymbol{\theta}),$$

where $\eta$ is the step size.

However, this supposition of taking a 'small step' in the gradient direction hides a specific assumption. In fact, it is equivalent to minimising a linear approximation (1st order Taylor expansion) of our loss function with an added penalization based upon the change in our parameters. Specifically, this is the optimum solution of

$$\delta\boldsymbol{\theta}^{\mathrm{GD}} = \operatorname*{argmin}_{\delta\boldsymbol{\theta}} \mathcal{L}(\boldsymbol{\theta}) + \nabla \mathcal{L}(\boldsymbol{\theta})^\top \delta\boldsymbol{\theta} + \frac{1}{2\eta} \delta\boldsymbol{\theta}^\top \delta\boldsymbol{\theta}.$$

where, $\delta\boldsymbol{\theta}^\top \delta\boldsymbol{\theta} = ||\delta\boldsymbol{\theta}||_2^2 = \sum_i \delta\boldsymbol{\theta}_i^2$. Note, one can derive gradient descent by simply finding the minimum of this optimization function (where the derivative with respect to $\delta\boldsymbol{\theta}$ is zero). Thus, gradient descent assumes that it is sensible to measure and limit the distance of our parameter update in terms of the squared (Euclidean) norm of the parameter change. Natural gradients supposes that this is not the best choice.

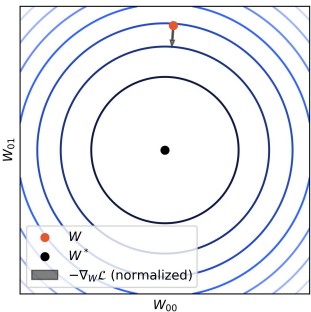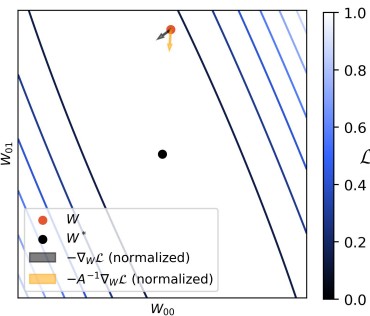

Figure 1: Left is shown a loss function landscape (in parameter space) which is a well-conditioned convex loss. In red is shown a model being optimised and the gradient descent direction (grey arrow) points toward the minimum of the function. Right, the loss function is modified to include a positive semi-definite scaling matrix ($\mathcal{L} = \sum_{(\boldsymbol{x},\boldsymbol{y}) \in \mathcal{X}} (\boldsymbol{W}\boldsymbol{x} - \boldsymbol{y})^{\top} \boldsymbol{A}(\boldsymbol{W}\boldsymbol{x} - \boldsymbol{y})$.) which causes a skew and stretch of the loss landscape. Gradient descent no longer points toward the minimum but instead in the direction of steepest descent (a detour), whereas natural gradient descent (orange arrow) points directly once more at the loss function minimum.

## 2.2 NATURAL GRADIENT DESCENT

Amari (1998), proposed the concept of natural gradients, and put forth the Riemann metric as the sensible alternative to a regular Euclidean measurement of distance. Specifically, rather than define squared distance with the Euclidean metric ($|\delta\boldsymbol{\theta}|^{\text{Euclidean}^2} = \sum_i \delta\boldsymbol{\theta}_i^2$), you can instead use a general metric instead,

$$|\delta\boldsymbol{\theta}|^{\text{Riemannian}} = \sum_i \sum_j g_{ij} \delta\boldsymbol{\theta}_i \delta\boldsymbol{\theta}_j = \delta\boldsymbol{\theta}^{\top} \boldsymbol{G}(\boldsymbol{\theta}) \delta\boldsymbol{\theta}$$

where $\boldsymbol{G}$ is the Riemann metric matrix. Note, that the values of the matrix, $\boldsymbol{G}$, are a function of $\boldsymbol{\theta}$ and are thus not static but depend upon $\boldsymbol{\theta}$. We can now make use of our Riemannian metric in place of the previous Euclidean metric such that,

$$\delta\boldsymbol{\theta}^{\text{NGD}} = \underset{\delta\boldsymbol{\theta}}{\text{argmin}} \, \mathcal{L}(\boldsymbol{\theta}) + \nabla\mathcal{L}(\boldsymbol{\theta})^{\top} \delta\boldsymbol{\theta} + \frac{1}{2\eta} \delta\boldsymbol{\theta}^{\top} \boldsymbol{G} \delta\boldsymbol{\theta}$$

and thus, by finding the minimum of this optimization (by the first derivative test), we can arrive at a neat formulation of natural gradient descent,

$$\delta\boldsymbol{\theta}^{\text{NGD}} = -\eta \boldsymbol{G}^{-1} \nabla\mathcal{L}(\boldsymbol{\theta}).$$

Natural gradient descent aims to ensure that it is not the steepest direction of descent which is taken, but instead the direction which undoes any skew in the loss function to arrive at a more direct descent toward (local) minima. This difference is illustrated in Figure 1.

One question remains, how might one arrive at a form for the Riemann metric matrix, $\boldsymbol{G}$?

### THE LOSS DISTANCE AS A PENALTY

Natural gradient approaches tend to find the form of this matrix by redefining models as probabilistic and thereafter forming a connection to the Fischer information matrix and information geometry (Amari, 1998; Martens, 2020). This unfortunately both obfuscates the intuition for natural gradients and disentangles it from deterministic (point) models with arbitrary losses.

We choose instead to describe this optimization in the deterministic regime and provide intuition of its impact, in a manner similar to that of Heskes (2000). Suppose that, instead of penalizing the Euclidean distance of our parameter change, that we instead penalize the distance traveled in *loss* space. Mathematically, we are supposing that

$$\delta\boldsymbol{\theta}^{\top} \boldsymbol{G} \delta\boldsymbol{\theta} :\approx \frac{1}{|\mathcal{D}|} \sum_{(\boldsymbol{x},\boldsymbol{y}) \in \mathcal{D}} (\ell(\boldsymbol{f}_{\boldsymbol{\theta}+\delta\boldsymbol{\theta}}(\boldsymbol{x}), \boldsymbol{y}) - \ell(\boldsymbol{f}_{\boldsymbol{\theta}}(\boldsymbol{x}), \boldsymbol{y}))^2.$$

Note, we here measure distance in terms of individual loss samples as otherwise one loss sample's value can increase while another's decreases - i.e. the total loss change can be degenerate to changes in the sample-wise losses.

It can be shown that by expanding this term with its Taylor series, see Appendix A, one arrives at $\boldsymbol{G} = \langle \nabla_{\boldsymbol{\theta}} \ell^\top \nabla_{\boldsymbol{\theta}} \ell \rangle_{x,y}$, bringing us to the same solution as found in general for the natural gradients learning rule update, where

$$\delta\boldsymbol{\theta}^{\mathrm{NG}} = -\eta \left\langle \nabla_{\boldsymbol{\theta}} \ell^\top \nabla_{\boldsymbol{\theta}} \ell \right\rangle_{x,y}^{-1} \nabla_\theta \mathcal{L}.$$

This intuition, that natural gradients can be viewed as a method for taking a step of optimization while regularizing the size/direction of the step in terms of loss difference, is a perspective which we believe is more interpretable while arriving at an equivalent solution.

## 2.3 NATURAL GRADIENT DESCENT FOR A DNN

Moving beyond the simple case of regression, we zoom into this problem when applied to multilayer neural networks. Defining a feed-forward neural network as

$$\boldsymbol{x}_i = \phi(\boldsymbol{h}_i) = \phi(\boldsymbol{W}_i \boldsymbol{x}_{i-1})$$

for $i \in [1...L]$, where $L$ is the number of layers in our network, $\phi$ is a non-linear activation function (which we shall assume is a fixed continous transfer function for all layers), $\boldsymbol{W}_i$ is a matrix of parameters for layer $i$, $\boldsymbol{x}_0$ is our input data, and $\boldsymbol{x}_L$ is our model output. We ignore biases for now as a simplification of our derivation.

Martens & Grosse (2015) (as well as Desjardins et al. (2015) and Bernacchia et al. (2018)) provided a derivation for this quantity. We demonstrate this derivation in Appendix B, and present its conclusions and assumptions here in short. Note that a first assumption is made here, that the natural gradient update can be computed for each layer independently, rather than for the whole network. This approximation not only works in practice (Desjardins et al., 2015) but is also fully theoretically justified for linear networks (Bernacchia et al., 2018). Taking only a single layer of a network, one may determine that

$$\boldsymbol{G}_{\boldsymbol{W}_i} = \langle \nabla_{\boldsymbol{\theta}_{\boldsymbol{W}_i}} \ell^\top \nabla_{\boldsymbol{\theta}_{\boldsymbol{W}_i}} \ell \rangle_{\boldsymbol{x}_0, \boldsymbol{y}} = \left\langle \mathrm{Vec}\left(\frac{\partial \ell}{\partial \boldsymbol{h}_i} \boldsymbol{x}_{i-1}^\top\right)^\top \mathrm{Vec}\left(\frac{\partial \ell}{\partial \boldsymbol{h}_i} \boldsymbol{x}_{i-1}^\top\right) \right\rangle_{\boldsymbol{x}_0, \boldsymbol{y}}$$

where the update is computed for a (flattened) vectorised set of parameters, indicated by the Vec() function.

After inversion, multiplication by the gradient, and reorganisation using the Kronecker mixed-product rule (with the additional assumption that the gradient signal is independent of the activation data distribution) one arrives at

$$\delta\boldsymbol{\theta}^{\mathrm{NG}}_{\boldsymbol{W}_i} = -\eta \mathrm{Vec}\left( \left\langle \frac{\partial \ell}{\partial \boldsymbol{h}_i}^\top \frac{\partial \ell}{\partial \boldsymbol{h}_i} \right\rangle_{\boldsymbol{y}}^{-1} \left\langle \frac{\partial \ell}{\partial \boldsymbol{h}_i} \boldsymbol{x}_{i-1}^\top \right\rangle_{\boldsymbol{y}, \boldsymbol{x}_{i-1}} \left\langle \boldsymbol{x}_{i-1}^\top \boldsymbol{x}_{i-1} \right\rangle_{\boldsymbol{x}_{i-1}}^{-1} \right). \tag{1}$$

Examining the terms of this update, we can see that the original gradient term (as you would compute by backpropagation) is in the middle of this equation. From the left it is multiplied the inverse correlation of the gradient vectors - tackling any skews in loss landscape as we visualized above. From the right the gradient descent update is multiplied by the inverse data correlation - specifically the data which acts as an input to this particular layer of the network. The presence of the inverse of the data correlation term is not only surprising but also an underappreciated aspect of natural gradients.

Note that we here refer to these outer product terms as correlations, despite the fact that a true correlation would require a centering of data (as would a covariance) and normalization. This is for ease of discussion.

## 2.4 THE ISSUE WITH DATA CORRELATIONS

In Figure 2, we visualise the impact of data correlations within a linear regression problem. Note that, as for Figure 1, we are visualising a loss landscape for a simple linear regression problem

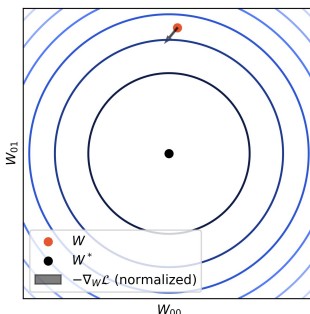 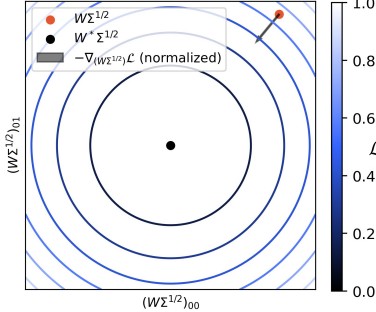

Figure 2: Left, the gradient descent vector is shown for a linear regression problem, though now the input data $x$ has a correlation structure ($\langle xx^\top \rangle = \Sigma$. Right, if the correlation matrix is integrated into the parameters, the alignment with direction of steepest descent in this landscape is returned.

(equivalent to a single layer network). As is clear, when our data itself has some correlation structure present, gradient descent once again points off-axis such that it would take a detour during optimization.

It is ultimately rather trivial to show why data correlations ruin gradient descent. If you consider a new set of data with some correlation structure, $\langle xx^\top \rangle = \Sigma$, you may equally (supposing well conditioned data) write this data as a whitened dataset multiplied by the matrix square root of the correlation matrix, $x = \Sigma^{1/2} \bar{x}$ where $\langle \bar{x}\bar{x}^\top \rangle = I$. As such, we can look at the output of a linear model of our inputs, $\hat{z} = Wx = (W\Sigma^{1/2})\bar{x}$, as a model in which this additional correlation matrix is a matrix by which our parameters are being brought into a non-orthonormal relationship. Thus, if we compute gradient descent with respect to our parameters, $W$, without accounting for the correlation inducing matrix, $\Sigma^{1/2}$, we have updated parameters which are no longer orthonormal, compare Figure 2 right.

This perspective, that input correlations at every layer of a deep neural network cause a non-orthonormal relationship between parameters, is our first major contribution. In the work which follows, we focus upon undoing data correlations and investigate how this impacts learning in neural networks.

## 3 SIMPLE DECORRELATION MECHANISMS CAN RID NETWORKS OF DATA CORRELATIONS

Above we have shown that data correlations impact the relationship between parameters at linear transformations. Thereby, the direction of gradient descent is skewed. We now go on to show how this can be corrected for.

There are multiple potential routes for correction of the gradient descent direction. One may measure correlation structure and directly invert and apply this inversion to the gradient updates to move toward a natural gradient descent update rule. However, the data being fed into one's parameters is still correlated and therefore continues to contribute to a non-orthonormal basis.

Alternatively, we here describe methods for modifying neural network models such that the neural outputs at each layer are decorrelated via some operation, no matter whether we are doing inference or training. Regular gradient descent in such a case is now closer in its update to natural gradients and furthermore our parameters can now relate in an orthogonal basis. This is explained in greater detail in Section 3.3.

Note that hereafter we make use of gradient descent to update models and attempt to remove data correlations within our models. Notably, we do not attempt to remove gradient correlations, the better known aspect of natural gradients (left most component of Equation 1). Thus performance is potentially left available via that route.

## 3.1 Existing methods for decorrelation

Data correlations can be removed from every layer of a deep neural network via a number of methods. These correlations must be undone continuously to keep up with changing correlation structures within deep neural networks while they are trained. Two main approaches exist to tackle the issue of removing data correlations: measurement of the correlation and inversion (via matrix decomposition) or a direct, continually updated, estimate of a matrix which can achieve a decorrelated outcome.

A number of works exist for directly taking the inverse-square-root of the correlation matrix of some data. Desjardins et al. (2015) did so by measuring the correlation matrix of data at every layer of a neural network (at pre-defined checkpoints) and thereafter carried out a matrix decomposition and an inverse (as did Luo (2017)). Bernacchia et al. (2018) did this at a minibatch-level, measuring correlations within a mini-batch and inverting these individually. Batch-normalization has also been extended to whitening and decorrelation for deep neural network training in a similar fashion (Huang et al., 2018; 2019), and this principle has been extended to much deeper networks, though also while stepping away from the theoretical framing of natural gradients. These works went so far as to apply decorrelation methods to extremely deep networks (101-layer) networks (Huang et al., 2018; 2019).

Few methods have considered iteratively computing a decorrelation matrix directly (i.e. without inversion or matrix decomposition) (Ahmad et al., 2022; Dalm et al., 2024; Sussillo & Abbott, 2009). Some methods optimise a matrix for decorrelation alongside the regular weight matrices in a neural network by construction of an appropriate loss function capturing how data should be modified for correlation reduction (Ahmad et al., 2022; Dalm et al., 2024). Other methods (Sussillo & Abbott, 2009) instead do not carry out any decorrelation within a network but instead store a disconnected matrix containing an interatively updated inverse correlation matrix and use this for parameter updating with the goal of achieving recursive least squares optimization. Regardless, these methods were thus far not theoretically linked to natural gradients.

From computational neuroscience however, a number of methods for dynamic and recurrent removal of data correlations have been proposed from the perspective of competitive learning and inhibitory control (Földiák, 1990; Pehlevan et al., 2015; Oja, 1989; Vogels et al., 2011). Földiák (1990); Pehlevan et al. (2015); Oja (1989); Vogels et al. (2011) describe, in work spanning almost two decades, a set of learning rules between nodes which, via linear recurrent dynamics, push neural activities toward decorrelated states at fixed points of these systems. The rules proposed by all four examples rely upon a simple updating scheme in which recurrent connections within populations of nodes are updated by an 'anti-hebbian' parameter update, in short with parameter gradients proportional to node-output correlations. They each, however, contribute a unique perspective on how such an update can be useful, from iterative learning of PCA dimensions (Földiák, 1990; Oja, 1989), through alternative subspace constructions such as multi-dimensional scaling (MDS) extraction (Pehlevan et al., 2015), all the way to an implementation in spiking neural networks which explains the development of real inhibitory synaptic connection structures in neurons (Vogels et al., 2011). Investigation into even more detailed methods to describe decorrelational inhibitory dynamics continues into contemporary work Lipshutz & Simoncelli (2024). These methods all provide neat and easily implementable dynamical systems for competition and decorrelation, however in contrast to the machine learning examples above, these methods have all been developed in the context of single layer networks, for unsupervised learning purposes, with little consideration for efficient implementation or application to gradient descent.

Here we present a method which bridges across all of the above, allowing fast and stable decorrelation while having two equivalent formulations: one via a single weight matrix multiplication, and another as the fixed-point of a recurrent system of dynamics.

## 3.2 A novel decorrelation mechanism

To move beyond existing decorrelation methods, we propose a method of decorrelation within a neural network with the following properties:

1. Learns to decorrelate consistently, regardless of the scale of the decorrelation matrix

2. Ensures that decorrelation does not reduce net activity in a layer

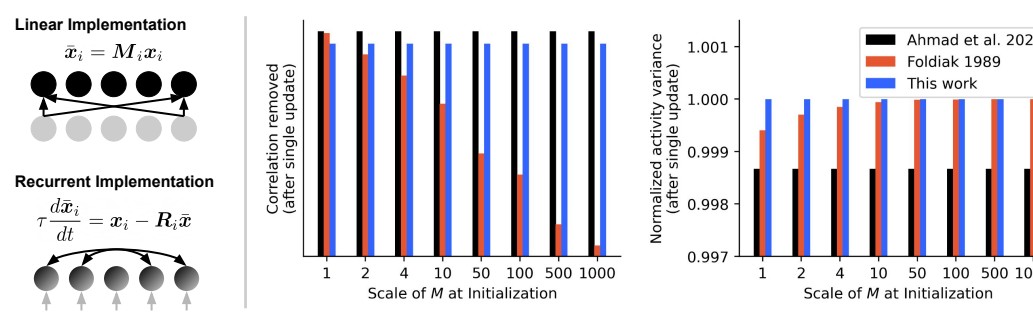

Figure 3: Left, our proposed method can be equivalently represented as either a single linear transformation or the fixed point of a recurrent set of dynamics (see the main text). Middle, existing methods for decorrelation through a system of dynamics (example of Földiák (1990) but representative of Vogels et al. (2011) and more) reduce activity correlations in a manner which is affected by the present scale (eigenvalues) of the decorrelating matrix. Right, existing methods also reduce the scale of all activity in a layer while decorrelating - tending toward the trivial solution of zero activity variance. The approach proposed in this work solves both of these issues.

3. Allows formulation of decorrelation as either an efficient linear transformation **or** equivalently via a distributed dynamical system

We propose that, at all layers of a neural network, one may construct a decorrelating transformation of the data. Efficiently, one can construct this as a linear transformation such that $\bar{\boldsymbol{x}}_i = \boldsymbol{M}_i \boldsymbol{x}_i$ where $\bar{\boldsymbol{x}}_i$ is intended to be a decorrelated form of the data $\boldsymbol{x}_i$. This means that a network now has one additional linear transformation per layer and the full network's computation is now written

$$\boldsymbol{x}_i = \phi(\boldsymbol{h}_i) = \phi(\boldsymbol{W}_i \bar{\boldsymbol{x}}_{i-1}) = \phi(\boldsymbol{W}_i \boldsymbol{M}_{i-1} \boldsymbol{x}_{i-1})$$

where all elements are as described previously, with the addition of a square decorrelating matrix $\boldsymbol{M}_i$ at every layer.

In order to learn this decorrelating matrix, one may update the decorrelating matrix in an iterative fashion such that

$$\boldsymbol{M}_i \leftarrow \boldsymbol{g}_i \circ (\boldsymbol{M}_i - \eta_M \langle \bar{\boldsymbol{x}}_i \bar{\boldsymbol{x}}_i^\top \rangle \boldsymbol{M}_i)$$

where $\eta_M$ is a learning rate for this update, and $\boldsymbol{g}_i$ is a scalar gain to ensures that each of the layer's decorrelated outputs has the same norm as it had prior to decorrelation at a layer or node level, $\boldsymbol{g}_i = \langle \boldsymbol{x}_i^2 \rangle / \langle \bar{\boldsymbol{x}}_i^2 \rangle$. Note that $\circ$ here indicates a Hadamard product (multiplication of each row of the matrix which follows). Note that we find results to be qualitatively independent of whether this normalization occurs at the node or layer-level, but the node-level description has less requirement for layer-wide information sharing and therefore greater biological plausibility.

This update minimises a loss capturing the correlations in $\bar{\boldsymbol{x}}$ in the style of Ahmad et al. (2022). Updates can be carried out in a stochastic fashion by simply measuring the correlations in $\bar{\boldsymbol{x}}_i$ within a minibatch for iterative updating, see Appendix E for the full pseudocode for such updating. Note that one can further improve upon the level of decorrelation by de-meaning the input data prior to this decorrelation process, such that $\hat{\boldsymbol{x}}_i = \boldsymbol{M}_i(\boldsymbol{x}_i - \boldsymbol{\mu}_i)$ where $\boldsymbol{\mu}_i$ is a unit-wise learned mean or batch-wise computed mean. We find this to further improve performance in practice.

One might enquire as to why this method is useful or interesting. Examining Figure 3, this decorrelation rule is effective at reducing the level of correlation at a network layer regardless of the existing scale (eigenvalues) of the decorrelating matrix - a problem faced by the existing learning rules of Földiák (1990), but also by the similar rules proposed by Pehlevan et al. (2015); Oja (1989); Vogels et al. (2011). Furthermore, this rule, via the gain factors $\boldsymbol{g}_i$, ensures that the scale (norm) of activities at any given layer remain at the existing scale, rather than reducing, see Figure 3 right. This ensures that decorrelating data does not tend towards the trivial decorrelation solution in which unit activities tend to zero.

Finally, aside from each of these benefits, this system can be easily converted to a system of recurrent dynamics with local-information, such that we may also arrive at a decorrelated state by defining,

$$\frac{d\bar{\boldsymbol{x}}_i}{dt} = \bar{\boldsymbol{x}}_i - \boldsymbol{R}_i\boldsymbol{x}_i,$$

with exact equivalence to our linear decorrelation matrix above, when $\boldsymbol{R}_i = \boldsymbol{M}_i^{-1}$. Notably, the matrix $\boldsymbol{R}_i$ can be locally updated to match $\boldsymbol{M}_i^{-1}$ via a Shermann-Morrison inverse computation. See Appendix D for a more complete description of such updating. This is particularly a benefit if one wishes to implement our proposed method for modelling of biological nervous systems or for application or translation of our models to analogue/neuromorphic devices.

One consideration to be made when adding any form of decorrelation (or alternative network modification) is it's additional computational complexity. Our linear transformation-based method for decorrelation adds a square decorrelating matrix for every layer of a neuron network of order. This translate to an additional matrix multiplication of order $\mathcal{O}(n_i^2)$ during inference at each layer, where $n_i$ are the number of nodes in layer $i$. Atop this, it adds a corresponding cost to training. Thus, for networks which are particularly wide (rather than narrow and deep), this can have a significant impact upon wall-clock execution time.

Our recurrent formulation requires numerical integration or other form of solving for states to reach their fixed point. Therefore its efficiency is highly dependent upon the exact software/hardware implementation and one should not consider it efficient for applications unless applied in a custom neuromorphic, analogue, or other exotic hardware solution (e.g. nervous systems). The exact computational cost of decorrelation is, for all of these reasons, highly dependent upon network architecture as well as implementation. Therefore it's utility must be examined on a case-by-case basis.

### 3.3 DECORRELATION BETTER ALIGNS GRADIENT DESCENT WITH NATURAL GRADIENTS

Having motivated and proposed our decorrelation methods, we here briefly demonstrate the impact that decorrelation of data has upon the natural gradients update.

As demonstrated in Section 2.3, the natural gradient update rule for a deep neural network can be expressed in the form of Equation 1. However, in the case in which decorrelation is successful, we have replaced states of layer $i$ such that $\boldsymbol{h}_i = \boldsymbol{W}_i\bar{\boldsymbol{x}}_{i-1} = \boldsymbol{W}_i\boldsymbol{M}_{i-1}\boldsymbol{x}_{i-1}$, and $\langle\bar{\boldsymbol{x}}_{i-1}\bar{\boldsymbol{x}}_{i-1}^\top\rangle = \text{diag}(\langle\bar{\boldsymbol{x}}_{i-1}^2\rangle)$. As such, the natural gradients update for a decorrelated input state are now computable as

$$\delta\boldsymbol{\theta}_{\boldsymbol{W}_i}^{\text{NG-decor}} = -\eta\text{Vec}\left(\left\langle\frac{\partial\ell}{\partial\boldsymbol{h}_i}^\top\frac{\partial\ell}{\partial\boldsymbol{h}_i}\right\rangle_{\boldsymbol{y}}^{-1}\left\langle\frac{\partial\ell}{\partial\boldsymbol{h}_i}\bar{\boldsymbol{x}}_{i-1}^\top\right\rangle_{\boldsymbol{y},\bar{\boldsymbol{x}}_{i-1}}\text{diag}(\langle\bar{\boldsymbol{x}}_{i-1}^2\rangle)^{-1}\right),$$

where now our gradient descent update (center) is now modified by a diagonal matrix via right multiplication rather than a full dense matrix. This diagonal matrix can no longer rotate the gradient vector, and thus only has a column-wise re-scaling effect upon the weight matrix update. See Appendix C for a discussion on why we focus on why we focus on decorrelation rather than whitening (which would make this term identity).

In this manner, decorrelation of input states at every layer of a network alleviates one half of the difference between the natural gradient update and regular gradient descent. This is particularly useful for application to algorithms which attempt approximate gradient descent as the gradient signal is compromised but the data signal is not. Therefore we cannot necessarily alleviate gradient correlations, as these are somewhat uncertain, but data correlations can be robustly removed to bring regular gradient descent closer to natural gradients.

## 4 APPROXIMATE METHODS FOR GRADIENT DESCENT WORK SIGNIFICANTLY BETTER WHEN COUPLED WITH DECORRELATION

The efficacy of decorrelation rules for improving the convergence speed of backpropagation trained algorithms is significant, as demonstrated in existing work (Huang et al., 2018; 2019; Dalm et al.,

2024). Rather than focus upon scaling this method to networks of significant depth (e.g. ResNet architectures as investigated by Huang et al. (2019) and Dalm et al. (2024)) we focus upon shallower networks and show how removal of data correlations enables existing approximate methods for backpropagation.

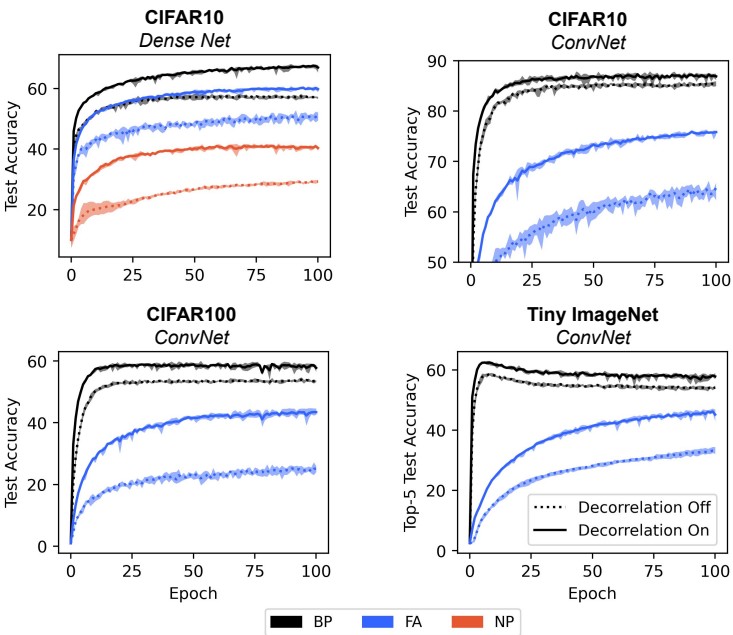

Figure 4: Adding a decorrelation mechanism at every layer of a neural network can massively speed up training convergence speed. Results are shown for backpropagation, feedback alignment, and node perturbation train upon the CIFAR10 in four hidden-layer dense networks, and for backpropagation and feedback alignment upon the CIFAR100 and TinyImageNet classification tasks in five hidden-layer convolutional neural networks. All learning methods were combined with the Adam optimizer (Kingma & Ba, 2014) and the categorical cross entropy loss. Network architectures and training hyperparameters are described in detail in Appendix F. Envelopes show the max and min accuracy levels across five randomly seeded networks.

It has been found that many existing 'biologically plausible' learning rules (i.e. ones which substitute backpropagation of error for alternative methods of gradient assignment which can be considered more plausible for distributed networks such as the brain) do not effectively scale to deeper networks and harder tasks (Bartunov et al., 2018). Recently, Dalm et al. (2023) made use of the decorrelation rule proposed by Ahmad et al. (2022) in order to enable training of multi-layer neural networks via the node-perturbation algorithm. In doing so, they did not draw a direct relation to the theories of natural gradients but were aided inadvertently by this effect.

Here we show that not only does decorrelation improve training of multi-layer neural networks with the node-perturbation algorithm (as has already been shown by Dalm et al. (2023)) but also significantly improves multi-layer network training when combined with implementation of the feedback alignment algorithm. Networks parameters in pseudocode, as well as the hyperparameters which were used for training are presented in Appendix F and an example of the computational pseudocode provided in Appendix E. Note that all simulations shown below are presented based upon a parameter grid search which was carried out individually for each credit assignment method and best parameters selected for each curve based upon a validation set.

Figure 4 shows the test-set performances of fully-connected and convolutional neural network models trained under various conditions and upon various tasks. As can be seen, backpropagation (BP) when coupled with decorrelation benefits from significant increases in generalization peformance and training speed in dense networks and smaller convolutional networks. Similarly, node perturbation (NP) is massively sped up, though suffers from a generally lower accuracy in this training regime.

Most significantly, the accuracy achieved by feedback alignment (FA) is increased far above what was previously achievable, even surpassing the accuracy of backpropagation in a dense network. For convolutional networks, the inclusion of decorrelation has a significant impact on peak performance and increases the speed of learning by orders of magnitude.

## 5 DISCUSSION

Herein we were able to link the concepts of natural gradients and decorrelation to show how, why, and to what degree decorrelating node activities at every layer of a neural network can massively boost performance of approximate gradient descent methods. The two approximate gradient descent algorithms treated herein are by no means the only algorithms which might be enabled by a decorrelation mechanism, and if one is interested in training of neural networks upon exotic hardware, one might consider combining such a distributed decorrelation mechanism with a number of alternative learning algorithms such as direct feedback alignment (Lillicrap et al., 2016; Nøkland, 2016), surrogate gradient learning Neftci et al. (2019) and many more. One must remain aware, however, that adding decorrelation to neural network architectures requires an additional weight matrix per neural network layer and induces additional computational overheads.

Outside of theoretical treatments and application spheres, decorrelation has been presented in neuroscientific work as an explanation of the filtering which happens at multiple levels of nervous systems. The center-surround processing which takes places at the earliest stage of visual processing (at retinal ganglion cells), for example, has been proposed as an aid for decorrelation of visual input to the brain (Pitkow & Meister, 2012). At a more general scale, inhibitory plasticity appears to be learned in a manner which also leads to spatial decorrelation (He & Cline, 2019), something which has also been modelled (Vogels et al., 2011). Thus it appears that decorrelation might be active in real nervous systems and could therefore aid in whatever form of optimization is taking place.

Beyond this, simulation work in computational neuroscience has also shown that via decorrelation of unit activities, competitive learning can be established to learn subspaces and carry out unsupervised feature extraction in a distributed and 'local' fashion (Bell & Sejnowski, 1997; Földiák, 1990; Oja, 1989; Zylberberg et al., 2011). Thus, it appears that decorrelational processes may also have utility unsupervised competitive learning approaches.

However, in real nervous systems nearby neurons can have significantly correlated activities - a feature which may be required for redundancy and robustness to cell death. Thus, as strict a decorrelational process as presented herein seems unlikely. Nonetheless, we point toward a promising method by which local and distributed learning can be enabled, and it remains to be investigated as to how this could be combined mapped more directly to real neural systems.

We find that our decorrelation approach has a combination of benefits: increased convergence speed (per epoch) along with increased generalization performance, most notably for BP. One question which we address only shortly in this work is the relative tradeoff of decorrelation vs whitening processes. Wadia et al. (2021) demonstrate that whitening approaches must be regularized to maintain generalization performance. Given our results, we propose that decorrelation might be a sensible tradeoff, where signal correlations are removed but remaining signal (or noise) is not excessively rescaled.

## 6 CONCLUSION

In this work we present an integration of a set of research directions including natural gradient descent, decorrelation and whitening, as well as approximate methods for gradient descent. Notably we illustrate and describe how one component of natural gradients is often overlooked and can be framed as correlations in feature data (at all layers of a neural network) bringing parameters into a non-orthonormal relationship. We show that data correlation at every layer of a neural network can be removed, in a similar fashion to neural competition, to enable orders of magnitude faster training. These results and insights together suggest that failures of 'biologically-plausible' learning approaches and learning rules for distributed computing can be overcome through decorrelation and the return of parameters to an orthogonal basis.

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

## A  Loss distance the approriate Riemann metric for natural gradients

In the main text, we propose that one may find the appropriate form of the Riemann metric for gradient descent by supposing that the metric measured should tend to the total summed sample-wise loss distance

$$\delta\theta^\top G\delta\theta \to \frac{1}{|\mathcal{D}|}\sum_{(x_0,y)\in\mathcal{D}}(\ell(f(\theta+\delta\theta,x_0),y)-\ell(f(\theta,x_0),y))^2.$$

By expanding the loss function with it's Taylor series,

$$(\ell(f(\theta+\delta\theta,x_0),y)-\ell(f(\theta,x_0),y))^2 = (\ell+\nabla_\theta\ell\delta\theta+\delta\theta^\top\nabla_\theta^2\ell\delta\theta+...-\ \ell)^2 \tag{2}$$

$$= (\nabla_\theta\ell\delta\theta+\delta\theta\nabla_\theta^2\ell\delta\theta+...)^2 \tag{3}$$

$$= \delta\theta^\top\nabla_\theta\ell^\top\nabla_\theta\ell\delta\theta+2(\nabla_\theta\ell\delta\theta\times\delta\theta^\top\nabla_\theta^2\ell\delta\theta)+... \tag{4}$$

$$\tag{5}$$

As the change in parameters tend to small values (or equivalently as $\lim_{\eta\to0}$), this term can be approximated by its first term. This first term, when used as the measure, is thus a simple equivalence to the Riemann metric

$$\delta\theta^\top G\delta\theta = \delta\theta^\top\left\langle\nabla_\theta\ell^\top\nabla_\theta\ell\right\rangle_{x_0\in\mathcal{D}}\delta\theta.$$

## B  Derivation for the Riemann metric for a deep neural network

Defining our model as

$$x_i = \phi(h_i) = \phi(W_i x_{i-1})$$

for $i\in[1...L]$, where $L$ is the number of layers in our network, $\phi$ is a non-linear activation function (which we shall assume is a fixed continuous transfer function for all layers), $W_i$ is a matrix of parameters for layer $i$, $x_0$ is our input data, and $x_L$ is our model output. We ignore biases for now as a simplification of our derivation.

First let us begin by collecting all parameter matrices into a single vector of parameters, such that $\theta=\text{Vec}(W_1,W_2,...W_L)^\top$ where $\text{Vec}(\cdot)$ indicates the flattening of a tensor into a vector. Next we can define, the derivative of our loss with respect to the parameters,

$$\nabla_\theta\mathcal{L} = \left(\frac{\partial\mathcal{L}}{\partial\text{Vec}(W_1,W_2,..W_L)}\right) = \left\langle\text{Vec}\left(\frac{\partial\ell}{\partial h_1}x_0^\top,\frac{\partial\ell}{\partial h_2}x_1^\top,...\frac{\partial\ell}{\partial h_L}x_{L-1}^\top\right)\right\rangle_{x_0\in\mathcal{D}}$$

where we have replaced the derivative terms with the layer computation as can be calculated by backpropagation. We can now compute more straightforwardly the outer product of this gradient vector with itself, such that

$$G(\theta) = \left\langle\nabla_\theta\ell^\top\nabla_\theta\ell\right\rangle_{x_0}$$

$$= \left\langle\text{Vec}\left(\frac{\partial\ell}{\partial h_1}x_0^\top,\frac{\partial\ell}{\partial h_2}x_1^\top,...\frac{\partial\ell}{\partial h_L}x_{L-1}^\top\right)^\top\text{Vec}\left(\frac{\partial\ell}{\partial h_1}x_0^\top,\frac{\partial\ell}{\partial h_2}x_1^\top,...\frac{\partial\ell}{\partial h_L}x_{L-1}^\top\right)\right\rangle_{x_0}$$

This is now a matrix of shape $G(\theta)\in\mathbb{R}^{M\times M}$ assuming $\theta\in\mathbb{R}^M$.

Rather than making explicit how to calculate the entire matrix, is is more fruitful to break down this computation into the separate blocks of this term, such that the $(i,j)$-block of our matrix is computed

$$G_{ij} = \left\langle\text{Vec}\left(\frac{\partial\ell}{\partial h_j}x_{j-1}^\top\right)^\top\text{Vec}\left(\frac{\partial\ell}{\partial h_i}x_{i-1}^\top\right)\right\rangle_{x_0} = \left\langle\left(x_{j-1}\otimes\frac{\partial\ell}{\partial h_j}\right)^\top\left(x_{i-1}\otimes\frac{\partial\ell}{\partial h_i}\right)\right\rangle_{x_0}$$

where $\otimes$ represents the Kronecker (Zehfuss) product. The mixed-product property of the Kronecker product allows us to also re-formulate this as

$$G_{ij} = \left\langle(x_{j-1}^\top x_{i-1})\otimes\left(\frac{\partial\ell}{\partial h_j}^\top\frac{\partial\ell}{\partial h_i}\right)\right\rangle_{x_0} = \left\langle x_{j-1}^\top x_{i-1}\right\rangle_{x_0}\otimes\left\langle\frac{\partial\ell}{\partial h_j}^\top\frac{\partial\ell}{\partial h_i}\right\rangle_y.$$

Note that we here made a separation of the expectation values based upon the fact that the loss value depends entirely on label or target output, $y$, and the network activities depend entirely on the value of the inputs, $x_0$. We assume, as Bernacchia et al. (2018) do, that these can therefore be separately averaged.

There are now two possible ways to proceed, each requiring an alternative assumption. First, one may suppose that we decide to optimise only a single weight matrix of the network at a time, which would allow us to ignore all but the diagonal blocks of our matrix, $G$. Bernacchia et al. (2018) alternatively arrived at a sole consideration of the (inverse) diagonal blocks of $G$ by assuming a linear network (i.e. that $\phi(x) = x$), and showing that in such a case, the matrix $G$ is singular and that it's pseudo-inverse can be taken using only the diagonal blocks.

Regardless, if we take a single block at a time, and suppose that we are only updating single weight matrices, we can now formulate the natural gradients learning rule as

$$\delta\theta_{W_i} = -\eta \langle \nabla_{\theta_{W_i}} \ell^\top \, \nabla_{\theta_{W_i}} \ell \rangle_{x_0}^{-1} \nabla_{\theta_{W_i}} \mathcal{L} \tag{6}$$

$$= -\eta \left( \langle x_{i-1}^\top x_{i-1} \rangle_{x_0} \otimes \left\langle \frac{\partial \ell}{\partial h_i}^\top \frac{\partial \ell}{\partial h_i} \right\rangle_y \right)^{-1} \nabla_{\theta_{W_i}} \mathcal{L} \tag{7}$$

$$= -\eta \left( \langle x_{i-1}^\top x_{i-1} \rangle_{x_0}^{-1} \otimes \left\langle \frac{\partial \ell}{\partial h_i}^\top \frac{\partial \ell}{\partial h_i} \right\rangle_y^{-1} \right) \text{Vec} \left\langle \frac{\partial \ell}{\partial h_i} x_{i-1}^\top \right\rangle_{x_0} \tag{8}$$

$$= -\eta \text{Vec} \left( \left\langle \frac{\partial \ell}{\partial h_i}^\top \frac{\partial \ell}{\partial h_i} \right\rangle_y^{-1} \left\langle \frac{\partial \ell}{\partial h_i} x_{i-1}^\top \right\rangle_{x_0} \langle x_{i-1}^\top x_{i-1} \rangle_{x_0}^{-1} \right) \tag{9}$$

$$\tag{10}$$

Thus, we arrive at an expression which allows us to better interpret the impact of computing the natural gradient. We can appreciate that the natural gradient formulation thus achieves two things: - The left matrix multiplication removes any skew and mis-scaling of the loss function with respect to the hidden activities, dealing with the problem that we classically associate with a skewed loss landscape - The right matrix multiplication removes the impact of any *correlation structure in our input data*. Correlation structure in our input-data is equivalent to our parameters living in a non-orthonormal basis set and thus affects the speed and accuracy of training!

## C  A NOTE ON DECORRELATION VS WHITENING

In this work we limit our emphasis on whitening and focus more upon decorrelation. The reason for this is two-fold. First, the off-diagonal elements of the decorrelation matrices cause the greatest impact in skewing gradient descent. The diagonal elements simply scale up/down the gradient vector and this can be trivially dealt with by modern optimizers (e.g. with the Adam optimizer (Kingma & Ba, 2014)). Second, for data in which features are often zero or extremely sparse, normalizing for unit variance can result in extremely large valued activations (distributions with extremely long tails) or unstable updates in the stochastic updating regime.

Wadia et al. (2021) pointed out that the restriction to whitened data can restrict the space of generalization unless properly regularized. This issue is one to keep in mind when developing methods which speed up training of models significantly and generalization performance should be a point of concern. However, as also described by Wadia et al. (2021), it may be that a regularized form of whitening/decorrelation would in fact be optimal for training for maximum generalization performance. Ultimately, we opt to avoid enforcing strict whitening and rely on decorrelation as an alternative and find it to be performative in practice.

# D OUR DECORRELATION MECHANISM AS A RECURRENT SYSTEM OF DYNAMICS

One may describe our proposed decorrelation mechanism as a system of lateral dynamics where $\frac{d\bar{x}}{dt} = \bar{x} - Rx$, where $R = M^{-1}$. This system can be shown to precisely arrive at the exact solution as outlined in the above text, $\bar{x} = Mx$. Furthermore, if one wished to update the parameters of this dynamic decorrelation setup in stochastic manner (single sample mini-batches), then one may apply the Sherman-Morrison formula to show that $M^{-1}$ should be updated with

$$M^{-1} \leftarrow (g \circ (I - \eta_M \bar{x}\bar{x}^\top)M)^{-1} = M^{-1}(I + \frac{\eta_M \bar{x}\bar{x}^\top}{1 - \eta_M \bar{x}^\top \bar{x}}) \circ g^{-1}$$

$$\approx M^{-1}(I + \eta_M \bar{x}\bar{x}^\top) \circ g^{-1}$$

$$\approx (M^{-1} + \eta_M(M^{-1}\bar{x})\bar{x}^\top) \circ g^{-1}.$$

After conversion to the notation with matrix, $R$,

$$R \leftarrow (R + \eta_M(R\bar{x})\bar{x}^\top) \circ g^{-1}.$$

Assuming that the total decorrelating signal to each individual unit can be locally measured ($R\bar{x}$) then this system of dynamics can also be updated in a local fashion by each node.

# E TRAINING PSEUDOCODE

---

**Algorithm 1** (Approximate) SGD with Decorrelation in a Multi-Layer Neural Network

---

**Input:** Input data $x_0$, Target Output $y$, Forward Weights $W_l$, Decorrelation Weights $M_l$, Total Number of Layers $L$, Forward Learning Rate $\eta_W$, Decorrelation Learning Rate $\eta_M$

```
Compute forward pass
```
**for** $i \in [0, 1, ..L - 1]$ **do**
    $\hat{x}_i = M_i(x_i - \mu_i)$            $\triangleright$ Demean and decorrelate state at layer $i$
    $x_{i+1} = \phi(W_{i+1}\hat{x}_i)$          $\triangleright$ Pass forward to the next layer
**end for**

```
Compute credit assignment method (Example BP)
```
**for** $i \in [L, L - 1, ...1]$ **do**
    **if** $i == L$ **then**
        $\delta_L = \phi'(x_L)(x_L - y)$     $\triangleright$ Compute gradient at output layer (regression or CCE)
    **else if  then**
        $\delta_i = \phi'(x_i)M_i^\top W_{i+1}^\top \delta_{i+1}$       $\triangleright$ Backpropagate (or replace $W_{i+1}$ for FA)
    **end if**
**end for**

```
Update Decorrelation Parameters
```
**for** $i \in [0, 1, ..L - 1]$ **do**
    $C = \hat{x}_i\hat{x}_i^\top$                $\triangleright$ Compute remaining correlation
    $g = \sqrt{x_i^2/\hat{x}_i^2}$              $\triangleright$ Compute rescaling
    $M_i \leftarrow g \circ (M_i - \eta_M C M_i)$     $\triangleright$ Update decorrelation matrix
    $\mu_i \leftarrow \mu_i + 0.1(x_i - \mu_i)$     $\triangleright$ Update mean estimate (fixed learning rate)
**end for**

```
Update Forward Parameters
```
**for** $i \in [0, 1, ..L - 1]$ **do**
    $W_{i+1} \leftarrow W_{i+1} - \eta_W \delta_{i+1}\hat{x}_i^\top$     $\triangleright$ Update by SGD (alternatively by other optimizer)
**end for**

---

Algorithm 1 describes pseudocode for the full computation and updating of a network which is carried out in this work. This is shown assuming the presentation of a single sample with its corresponding network update, though is in-practice used with a mini-batch of size 256 by default. The

algorithm is shown for the case of backpropagation, though it is compatible with alternative credit assignment methods such as feedback alignment or node perturbation. Furthermore, the pseudocode shows the forward parameters as being updated by SGD, though in practice these are updated with the Adam (Kingma & Ba, 2014) optimizer.

# F ARCHITECTURES AND HYPERPARAMETERS

Table 1: The neural network architectures for the experiments in Figure 4. For Convolutional and Pooling layers, shapes are organised as 'Kernel Width $\times$ Kernel Height $\times$ Output Channels (Stride, Padding)'.

| NETWORK | LAYER TYPES | LAYER SIZE |
|---|---|---|
| DENSE ARCHITECTURE | INPUT | $32 \times 32 \times 3$ |
| | FC | 1000 |
| | FC | 1000 |
| | FC | 1000 |
| | FC | 1000 |
| | FC | 10 |
| CONVOLUTIONAL ARCHITECTURE (CIFAR10/100) | INPUT | $32 \times 32 \times 3$ |
| | CONV | $3 \times 3 \times 32, (1, 0)$ |
| | CONV | $3 \times 3 \times 32, (1, 0)$ |
| | MAXPOOL | $2 \times 2 (2, 0)$ |
| | CONV | $3 \times 3 \times 64, (1, 0)$ |
| | CONV | $3 \times 3 \times 64, (1, 0)$ |
| | MAXPOOL | $2 \times 2 (2, 0)$ |
| | FC | 1000 |
| | FC | 10 (OR 100 FOR CIFAR100) |
| CONVOLUTIONAL ARCHITECTURE (TINY IMAGENET) | INPUT | $56 \times 56 \times 3$ |
| | CONV | $3 \times 3 \times 32, (1, 0)$ |
| | CONV | $3 \times 3 \times 32, (1, 0)$ |
| | MAXPOOL | $2 \times 2 (2, 0)$ |
| | CONV | $3 \times 3 \times 64, (1, 0)$ |
| | CONV | $3 \times 3 \times 64, (1, 0)$ |
| | MAXPOOL | $2 \times 2 (2, 0)$ |
| | FC | 1000 |
| | FC | 200 |

The specific networks trained for demonstration are of two types: a fully connected network architecture and a convolutional network architecture. The structures are show in Table 1. Two datasets are used for training and testing. These include the CIFAR10 and CIFAR100 datasets Krizhevsky (2009) and the Tiny ImageNet dataset derived from the ILSVRC (ImageNet) dataset Russakovsky et al. (2015).

The CIFAR10(100) dataset is composed of a total training set of 50,000 samples of images (split into either 10 or 100 classes) and 10,000 test images. The TinyImageNet dataset is a sub-sampling of the ILSVRC (ImageNet) dataset, composed of 100,000 training images and 10,000 test images of 200 unique classes. Images from the ILSVRC dataset have been downsampled to $64 \times 64 \times 3$ pixels and during our training and testing we further crop these images to $56 \times 56 \times 3$ pixels. Cropping is carried out randomly during training and in a center-crop for testing.

The training hyper-parameters were largely fixed across simulations, with a fixed (mini)batch size of 256, and Adam optimiser parameters of $\beta_1 = 0.9$, $\beta_2 = 0.999$, $\epsilon = 1e - 8$. Aside from these parameters, the learning rates for each simulation curve in Figure 4 were individually optimised. A single run of each simulation (curve), with a 10,000 sample validation set extracted from the training data, was run across a range of learning rates for both the forward and decorrelation optimisation independently. Learning rates were tested from the set $[1e-2, 1e-3, 1e-4, 1e-5, 1e-6, 1e-7]$ for

Table 2: The learning rates as selected for simulations shown in Figure 4.

| TRAINING ALGORITHM | HYPER PARAM | CIFAR 10 (DENSE) | CIFAR 10 (CONV) | CIFAR 100 (CONV) | TINY IMAGENET (CONV) |
|---|---|---|---|---|---|
| BP | ADAM LR | 1E-4 | 1E-3 | 1E-3 | 1E-3 |
| BP + DECORRELATION | ADAM LR | 1E-4 | 1E-3 | 1E-3 | 1E-3 |
| | DECOR LR | 1E-5 | 1E-5 | 1E-5 | 1E-5 |
| FA | ADAM LR | 1E-4 | 1E-4 | 1E-4 | 1E-5 |
| FA + DECORRELATION | ADAM LR | 1E-4 | 1E-4 | 1E-4 | 1E-5 |
| | DECOR LR | 1E-6 | 1E-5 | 1E-5 | 1E-5 |
| NP | ADAM LR | 1E-5 | - | - | - |
| NP + DECORRELATION | ADAM LR | 1E-4 | - | - | - |
| | DECOR LR | 1E-6 | - | - | - |

the forward learning rates, with and without decorrelation learning (again with learning rates tested from this set). Best parameters were selected for each simulation and thereafter the final results plots created based upon the full training and test sets and with five randomly seeded network models for each curve (see min and max performance as the envelopes shown in Figure 4.

