# OpenReview forum: "Correlations Are Ruining Your Gradient Descent"
_ICLR.cc/2025/Conference — Submitted to ICLR 2025_

### Official Review · Reviewer_HgPq · 2024-11-03

**Soundness:** 3
**Presentation:** 4
**Contribution:** 3
**Rating:** 5
**Confidence:** 3

**Summary:**

This submission didactically draws together approximate gradient methods, natural gradient descent, and decorrelation methods, to ultimately improve approximate gradient methods. More specifically the paper shows how natural gradients defined with respect to changes in the loss function highlight the importance of data correlation in determining the alignment between GD and natural gradient updates. With this insight in hand, approximate gradient descent methods are shown to be improved by decorrelating the network activations.

**Strengths:**

I believe this is an original synthesis of concepts resulting in an original and significant modification to approximate gradient methods.
This paper is extremely well written, and generally of high clarity and quality throughout.

**Weaknesses:**

- It is not clear that the alignment to natural gradients is what is underlying the improved performance.
- No assessment of how successful the decorrelation updates are in aligning regular updates to the natural updates. How big is the contribution of gradient correlations?
- While the improvements in FA and NP are striking, the experimental results are not finely tuned. It is possible with GD to obtain better results than those shown, and it is unknown if FA and NP will match this improvement.
- To verify decorrelated FA scales, harder experiments would strengthen the paper (e.g those in Bartunov 2018. Assessing the Scalability of Biologically-Motivated Deep Learning Algorithms and Architectures for ANNs.)
- Given biological relevance, showing application and relevance to recurrent networks would improve the paper.

**Questions:**

- typo optimization 153
- Assumption on 182 that NGD independent layers, how serious is this?
- 163, should the expectations on G not be over y too?
- 251 skewed is being used non-technically?
- 260 type orthogonal
- 264 citation missing, gradient correlations
- What is the connection between natural gradients and the approximate gradient methods. Why in particular should the approximate methods benefit from natural gradients?
- The discussion of the orthonormal basis of parameters (page 5) induced by data correlations does not make sense to me. Are the parameters already not orthornormal?

---

> ### Author Response · Authors · 2024-11-15
> **Review Acknowledgement and Rebuttal**
>
> First, thank you for your feedback. We begin below by providing a general rebuttal, followed by a numbered response to each point made in the weaknesses and questions sections of your review. Note that we have also uploaded a revised copy of the manuscript.
>
> It appears that a number of aspects of our work were mis-understood or unclear. We see this as something we can also correct by modification of our paper and therefore we have added Section 3.3 to demonstrate the exact benefit of decorrelation for aligning to natural gradients and express why this is important for approximate gradient descent methods.
>
> To give our reviewer more clarity: At its core, our work demonstrates that regular gradient descent has two core properties which are corrected by natural gradient descent when updating linear transformation (matrix multiplications). One being correlation structures in the gradient signal, and the other being correlation structures in input data distributions. We demonstrate that input correlation structures put parameters of weight matrices into a non-orthonormal relationship: i.e. that correlated inputs are equivalent to weights embedded in a correlated basis space. For approximate gradient descent methods: we cannot improve the gradient signal (it is inherently uncertain) but we can remove the impact of input correlations upon weight updates by carrying out a decorrelation procedure. We try to make this clearer in our work by the addition of Section 3.3 and general improvements to the flow and structure of the paper (as requested also by our other reviewers).
>
> We hope that this clears up general mis-understandings which were present in this review. As the authors, we are curious as to what additional aspects of the paper were deemed to be of low quality such that the rating was below acceptance? Any further clarity would be appreciated.

---

> > ### Author Response · Authors · 2024-11-15
> > **Point-by-point rebuttals**
> >
> > #### _Weaknesses Rebuttals:_
> > 1. We hope that our general rebuttal above and the newly added Section 3.3 make the relationship between natural gradient descent, decorrelation, and approximate gradient descent clearer.
> > 2. Ultimately, it is difficult to determine how much of an impact gradient correlations have versus data correlations. This is entirely dependent upon the proposed task and downstream network. However, for the case of approximate gradient descent, we already assume that we do not have access to perfect gradient information (due to the approximate nature) and therefore the gradient correlation contributions could not (even in theory) be accurately measured. Section 3.3 has been amended to add this explanation.
> > 3. This point appears to be a misunderstanding: The experimental results were in fact tuned via a hyperparameter grid-search. This grid search was carried out for each algorithm (BP/FA/NP) independently and the best hyperparameters for each algorithm selected. The results for BP are not at SOTA levels as these networks are rather straight-forward convolutional networks without many bells and whistles (e.g. no dropout or additional regularizations). This was in order to ensure that the benefits observed can be isolated to only interactions between the decorrelation and learning rules.
> > 4. One of the reasons to select tiny ImageNet (200 classes from ImageNet with downsampled images) was to select a sufficiently difficult task to stretch our theory beyond CIFAR10/100. Bartunov et al. tested many modified networks with custom local convolutional implementations, code for which is however unfortunately unavailable. What would satisfy our reviewer in terms of network/task? Is a full ImageNet run the desired result in this case?
> > 5. We do agree that recurrent networks are interesting - in fact we do further expand on our dynamical systems implementation of decorrelation in Figure 2. Is there a particular task or recurrent network setup which this reviewer would like to see? Application of the decorrelation methods to more machine learning-esque networks (without continuous time dynamics) is difficult to reconcile with biology so this is rather unclear, as are approximate gradient methods for recurrent neural networks.
> >
> > #### _Questions Rebuttals:_
> > 1. Typo fixed, thank you!
> > 2. It is unclear how impactful this assumption is. Empirically, it seems that considering layer-wise correlation structures is effective, and whole-network correlation structures are extremely expensive to compute. Therefore this appears to be a sensible scale on which to operate in practice. Theoretically, this entirely depends upon the level of data and gradient correlation between layers of a network.
> > 3. Correct, an expectation over y is also being taken - this has been corrected in all equations.
> > 4. Skewed is indeed being used non-technically, sheared would be more accurate but unless our reviewer has an issue with this, we find skewed to be a reasonable way to communicate this.
> > 5. Typo corrected, thank you.
> > 6. Here we are referring to our earlier natural gradients derivations and have made this clearer by reference to equation 1.
> > 7. Please see the above rebuttal as well as Section 3.3 for clarification on the impact of decorrelation on natural gradients and thereby upon approximate gradient descent methods.
> > 8. Parameters in general are simply values operating on some input data. In that sense, these parameters act in an orthonormal fashion only if the input data itself has no correlation structure. Otherwise, one may equivalently write the parameters as having correlation structure and inputs as having none. This is one key conceptual contribution which we provide and is perhaps explained best in Figure 2 and it’s accompanying text.
> >
> > We thank the reviewer for their keen attention to our text. We hope that, if our changes are sufficient, that you might consider a modification of your grade. If there are any remaining questions, we would be happy to hear them and to hear feedback on specific examples for improving on weakness points 4 and 5.

---

> > > ### Comment · Reviewer_HgPq · 2024-11-19
> > >
> > > I thank the authors for their response and added clarifications. They have cleared up the majority of my questions.
> > >
> > > *For approximate gradient descent methods: we cannot improve the gradient signal (it is inherently uncertain) but we can remove the impact of input correlations upon weight updates by carrying out a decorrelation procedure.*
> > >
> > > Thank you for this clarification. This does make the connection to approximate methods clearer, ie. that input decorrelation is well motivated for these approaches. However, there is still no link to why approximate methods would benefit in particular from decorrelation vs standard gradient descent. Should both not benefit equally from decorrelation?
> > >
> > > *We demonstrate that input correlation structures put parameters of weight matrices into a non-orthonormal relationship: i.e. that correlated inputs are equivalent to weights embedded in a correlated basis space.*
> > >
> > > Thank you for this clarification.
> > >
> > > I still think the experimental results are a weak point of the submission. In my opinion these results are an important demonstration of the preceding theory and are what elevates the work beyond more of a perspective/review (similar concerns to reviewer qqzv). Regarding weakness points 3 & 4, my point here is that many approximate gradient approaches have been shown to work in simple settings, but failed when put to harder tests. For relevance and impact to the field it is therefore important to show empirical results demonstrating improved performance of e.g. FA over previously published results - as this is a known issue with approximate methods. For example, Bartunov reports similar performance for FA on BP on cifar10 but markedly different for imagenet. First, relatively simple experiments with resnets using sgd, momentum and weight decay can significantly improve over the results presented for cifar10 and 100. Regarding imagenet runs, while I do not think they should be required per se, they would undoubtedly strengthen the paper - as previous results show imagenet is where approximate methods fall down. The tiny imagenet experiments are encouraging and I would urge the authors to run imagenet, and with libraries such as ffcv and a100s such experiments can take less than a day. At a minimum, the experiments should be run for longer, 200-300 epochs, and tiny imagenet is obviously not converged.
> > >
> > > Regarding RNNs. I do not have a particular experiment in mind. But for example eprop (Bellec 2020) (or an alternative https://www.jmlr.org/papers/v21/19-562.html) on a simple task (e.g. row sequential mnist or the penn treebank dataset) could demonstrate application to recurrent networks.

---

> > > > ### Author Response · Authors · 2024-11-25
> > > >
> > > > Thank you for your continued engagement with our work. Some clarifications and input are given below.
> > > >
> > > > - `However, there is still no link to why approximate methods would benefit in particular from decorrelation vs standard gradient descent. Should both not benefit equally from decorrelation?`
> > > > Indeed BP also benefits majorly from decorrelation (see Figure 4). The benefit of decorrelation is present for both backpropagation and alternative algorithms. The specific benefit for BP has also been shown in existing work by Dalm et al. 2024 “Efficient Deep Learning”. However, these existing work could never explain _why_ decorrelation was useful. This work bridges that gap by giving an explanation of why decorrelation works at all. Furthermore, decorelation may be an interesting tool or addition for BP but has a much bigger implication for biologically plausible algorithms which traditionally underperform. For bio-plausible algorithms, decorrelation is a straightforward addition which can massively boost performance and has real neural correlates that can justify its presence.
> > > > - `I still think the experimental results are a weak point of the submission.` We understand the concern and perhaps only differ in opinion on whether this current work meets the level of contribution for acceptance. To be clear: although BP-based networks (with well known hyperparameter tunings) can be trained within a day on imagenet, FA networks have a custom layer-type and require hyperparameter tuning: therefore take much longer to simulate per epoch but also to find the right hyperparameters. Furthermore, adding decorrelation increases computational expense further and introduces additional hyperparameters. In this respect, parameter sweeping for even TinyImageNet was a computationally expensive undertaking. If this work is rejected on grounds of simulation scale, then we shall have to consider that our opinion on the scale should be updated.

---

### Official Review · Reviewer_qqzv · 2024-11-03

**Soundness:** 3
**Presentation:** 2
**Contribution:** 2
**Rating:** 3
**Confidence:** 3

**Summary:**

In this work, the authors address the challenge of data correlations impacting neural activity and the learning process within neural networks. They take a comprehensive approach, beginning with an analysis that connects data correlations to the non-Euclidean geometry of the parameter space. Specifically, they demonstrate that correlations in input data lead to longer paths through parameter space during learning. This observation is rooted in the theory of Natural Gradient Descent, as originally developed by Amari in the late 90's, where parameter space is considered curved and defined locally by the inverse Fisher information matrix. The authors provide evidence that input correlations influence these longer paths.
Next, they propose an incremental learning rule capable of decorrelating inputs within each neural network layer. This rule facilitates a linear transformation to reduce input correlations progressively. The authors demonstrate that employing such a decorrelation mechanism accelerates learning and potentially improves generalization across several learning paradigms, including standard backpropagation, Node Perturbation, and Feedback Alignment. The concluding part of the paper discusses the implications of removing correlations in biological and artificial settings.

**Strengths:**

• The initial sections of the paper provide an insightful overview of Natural Gradient Descent and offer a compelling perspective on how Amari’s theoretical framework can be contextualized within deep learning. While typical applications of Natural Gradients focus on Bayesian inference, where the Fisher information naturally defines the metric, the authors present an innovative interpretation that broadens its relevance.

• The simulations included in the study convincingly demonstrate that implementing decorrelation at each layer significantly enhances training efficiency, particularly within the Feedback Alignment framework.

• The paper offers a comprehensive overview of the impact of data correlations on training dynamics, which serves to enrich the reader’s understanding of this often underexplored aspect of deep learning optimization.

**Weaknesses:**

The primary issue with this paper is its limited coherence and lack of substantial innovation. The manuscript presents small incremental contributions across several topics and attempts to integrate them into a unified framework. This approach results in a paper that straddles the line between a review and an opinion piece, which does not align well with the expectations for an ICLR submission. Below, I outline the specific areas where the paper falls short in terms of novelty:
1. **Review of Natural Gradient Descent**: The first section provides a summary of Natural Gradient Descent, highlighting that input correlations can be inferred from the derivation of the metric tensor. However, this observation is neither surprising nor particularly useful, as it merely reiterates that input correlations are intrinsic to Natural Gradient theory. Moreover, the concept of learning dynamics in curved parameter spaces is not applied or built upon in subsequent sections, limiting the relevance of this part of the paper to the overall narrative.

2. **Derivation of a Local Learning Rule**: The second section presents a derivation for a local learning rule intended to decorrelate inputs to neural network layers. While the authors point out the novelty of their incremental rule (line 340), this rule closely resembles existing methods such as recursive least squares (e.g., Sussillo and Abbott, 2009), which also stem from the goal of decorrelating neural activity. The resemblance is even clearer when studying the derivation of the proposed decorrelation learning rule in Appendix C. Notably, the proposed rule introduces a novel scalar rescaling that maintains the norm. However, this addition is minor, and the overall innovation is minimal.

3. **Numerical Experiments on Layer-wise Decorrelation**: The third section includes numerical results showing the advantages of input decorrelation for training deep networks. Prior work has already demonstrated that input decorrelation accelerates learning in frameworks such as backpropagation and Node Perturbation. Consequently, the main novelty here is the improved performance observed in the Feedback Alignment framework. However, the paper does not provide sufficient explanation or analysis to elucidate why decorrelation significantly enhances Feedback Alignment, nor does the performance match standard backpropagation. This omission limits the impact of this result, rendering it another incremental contribution.

Collectively, these points suggest that the novelty presented in this work does not meet the standards expected for ICLR.

Finally, the discussion section of the paper, while extensive, lacks coherence and leans heavily on speculation. For instance, the portion discussing biological plausibility is unclear in its message. Although the authors correctly note that decorrelation is observed in various neural circuits, the connection between these observations and the mechanisms or theories presented in the paper remains ambiguous. Additionally, the claim that decorrelation improves generalization (line 479) is inadequately supported by the data. The training curves in Figure 4 do not appear to reach convergence, as indicated by the non-saturating test accuracy. This raises doubts about the robustness of the claimed generalization improvements.

**Questions:**

To better align this work with the standards and expectations of an ICLR paper, the authors need to sharpen their message and clearly articulate the novel contributions of each section. Specifically, the authors may consider addressing the following questions:
1. How does the consideration of Natural Gradient Descent and the dynamics in curved parameter space contribute to the understanding or derivation of the incremental learning rule? Can this rule be directly derived from the principles of the Natural Gradient theorem?
2. What differentiates the proposed learning rule from existing frameworks for decorrelating neural activity, such as Recursive Least Squares? If there are differences, what are the specific theoretical or practical advantages of the proposed rule?
3. Could the authors provide more extensive numerical analyses to demonstrate whether the benefits of their approach extend beyond faster learning to improved generalization? While Natural Gradient theory primarily explains accelerated convergence, do the authors observe any substantial gains in generalization performance?

Lastly, I want to note that I enjoyed reading the paper and its broad review, which connects the concept of natural gradients to deep network training. However, this discussion currently feels more suited to an opinion piece.

---

> ### Author Response · Authors · 2024-11-15
> **Review Acknowledgement and Rebuttal**
>
> First, thank you for your in-depth analysis and thinking along with our work. To best answer your concerns and questions, we first give an overall rebuttal followed by a list of points which each address your core questions/weaknesses (where 1, 2, and 3 are responses to both the weaknesses and questions 1, 2, 3 combined). Note that we have also uploaded a revised copy of the manuscript.
>
> It appears that there is some concern that our work is more suited to a review or opinion and has little novelty. We see this rather differently. Our paper aims to bring synthesis of multiple ideas together as well as providing theoretical and empirical results and is not solely an opinion paper in that regard. First, there is novel theory in describing non-orthonormal parameter relations which are induced by correlations (and related to natural gradients), second a multi-form decorrelation rule which has both an efficient linear transformation form as well as a dynamical systems form without the drawbacks of existing decorrelation methods, and third entirely novel results which demonstrate the efficacy of the preceding theory. On top of this, we bring together three topics which have not been discussed in conjunction in existing literature; natural gradients, decorrelation, and their impact upon approximate gradient descent algorithms.
>
> We believe that contributions such as ours are even more valuable in a time where the number of narrowly-focussed papers are growing exponentially while papers providing deep understanding and synthesis are lacking in number. We also believe that existing researchers developing approximate gradient descent methods in this field will appreciate and build atop this work. An appreciation which is even demonstrated by the positive review by one of our other reviewers.
>
> However, we do see that our story-telling could perhaps be polished somewhat. In service of that, and in service of answering your concerns we present some revisions and perspectives below.
>
> #### _Weaknesses/Question Rebuttals_
>
> 1. The relation of natural gradients to non-orthonormality of parameter space which is induced by correlations is the core contribution which is not provided in existing literature. However, we did not, in our first submission, convincingly relate our contribution on natural gradients to those of decorrelation. Therefore, we attempt to clear this up in the text by the addition of Section 3.3 where our decorrelation development is directly linked to the natural gradients update rule and implications discussed. The impact is precisely upon a removal of one term (the correlation term) which may otherwise act to rotate the gradient descent vector away from a more optimal direction.
> 2. We appreciate that our discussion on existing decorrelation learning rules may not have been as detailed as desired by this (and other) reviewers. We have therefore extended this section to discuss to a greater extent the existing work in computational neuroscience (see Section 3.2, L295-L313). I would ask this reviewer to note that the RLS rule proposed by Sussilo et al. in fact is not applied in the same manner as we do here - they in fact propose an introduction of a new weight matrix multiplication outside the dynamics of the nodes which aids in carrying out RLS and this is a biologically implausible addition. Our proposal is to bring decorrelation into network dynamics so that updates can be carried out ‘as if’ one were carrying out GD. Furthermore, Sussillo et al do not provide a dynamical systems implementation of their rule, perhaps precisely because it is not embedded within nodes. Nonetheless, we appreciate that the work by Sussillo also inverts the data correlation matrix to achieve RLS and we additionally refer to this in section 3.2 but make clear that this is outside of the 'activity-space’ of a network with the text:
> `Other methods \citep{Sussillo2009-ws}, instead do not carry out any decorrelation within a network but instead store a disconnected matrix containing an interatively updated inverse correlation matrix and use this for parameter updating with the goal of achieving recursive least squares optimisation.`
> 3. For point three, we would request some additional input. From our perspective, Figure 4 (especially for BP but also for FA) robustly demonstrates that there is a strong generalisation benefit by the addition of decorrelation. In fact, regular BP simulations begin overfitting before the end of almost all simulations, showing their generalisation limit. Is there something unclear in this conclusion? Do you believe that additional input would be useful?
>
> We do believe these modifications have improved our work, and we thank you. We also hope that, if they are sufficient, that you would consider a modification of your grade.
>
> Are there other specific changes or clarifications that this reviewer would request?

---

> > ### Comment · Reviewer_qqzv · 2024-11-22
> >
> > Thank you for the response and carefully considering my comments. However, I am still not convinced that this work reaches a high level of novel contribution, per the ICLR guidance.
> >
> > In your response, you wrote:
> >
> > > Contributions such as ours are even more valuable in a time where the number of narrowly-focussed papers are growing exponentially while papers providing deep understanding and synthesis are lacking in number.”
> >
> > I agree that broad perspectives are needed. However, I don’t feel like this paper gave me a deep understanding. You point out interesting relations between natural gradients and correlations, and you have convinced me there may be unexplored benefits in considering natural gradients. However, while interesting, this idea is not well grounded, and the paper feels like tying several results together. My main concerns about the contribution and coherence of this paper still hold.
> >
> > Regarding your specific responses, here are my more elaborate responses:
> >
> > 1. The addition of section 3.3 did not help me better understand the connection between natural gradients and the decorrelation step. It feels like an afterthought. The natural gradients do not help me understand the decorrelation better, and deriving the decorrelation does not require natural gradients.
> >
> > 2. Thank you for the clarification regarding the decorrelating learning rule. I still believe it resembles RLS, but I agree that you offer a new online version. However, your decorrelating learning rule is still nearly identical to [Dalm et al. 2024 “Eddective Learning With Node Perturbation”], which you quote. The main difference is that you subtract the diagonal term in updating the decorrelation term. Unless there is a central and conceptual benefit for this extra term, I struggle to see the novelty.
> >
> > 3. The slight improvement of BP and node perturbation due to input decorrelation was demonstrated in previous studies, which you cite (Dalm et al. 2024 “Effective Learning With Node Perturbation” and Dalm et al. 2024 “Efficient Deep Learning”). Thus, the novel contribution here uses decorrelation methods similar to those proposed before for FA. Furthermore, I don’t see how these results relate to the natural gradients. As I mentioned above, and as noted by the previous papers that you cite, there is no need for natural gradients in applying these decorrelating steps. Thus, it adds to the feeling that the paper brings together several results and does not tie them well.

---

> > > ### Author Response · Authors · 2024-11-25
> > >
> > > Thank you for your thoughts. They are genuinely useful for us to iterate this paper well.
> > >
> > > 1. This point is similar to your general perspective (that `I don’t feel like this paper gave me a deep understanding`) Perhaps the missing link in this case is that we intend to give a clear theoretical explanation of why decorrelation helps in training convergence speed for BP and why it helps so much for bio-plausible algorithms. Therefore, our explanation of natural gradients is actually presented to give an insight in this regard: that decorrelation is helpful because it removes half of the difference between natural gradient descent and regular gradient descent. That it relieves the parameter non-orthonormality that is present when linear layers process correlated data. If there are particular ways in which you would recommend that we re-write the paper to make this more explicit, we would appreciate input. However, at present we believe that we are making this point well and it is not clear how to make this point more explicit.
> > >
> > > 2. The learning rule in our work is very nearly the same as Dalm et al (2024) but actually due to a common origin in Ahmad et al. (2023) (Constrained Parameter Inference as a Principle for Learning). And we in fact compare against this rule directly in Figure 3. Indeed the main difference is a missing diagonal term and a rescaling. However, this difference is crucial for re-writing the decorrelation rules as a system of dynamics. With the original diagonal term, one cannot make use of the Shermann-Morrison inverse (see Appendix D). This is because one cannot reduce the original rule of Ahmad et al. (2023) to a rank-1 update. We hope that this clarifies this point further.
> > >
> > > 3. This point is well taken, but i believe misses the key theoretical contributions of this paper. No previous works that you have listed give a theoretical explanation of __why__ decorrelation works. Only that it does. This work aims to bring theoretical rigour to all of the mentioned works and also to show that the benefits do not exist only for specific algorithms but for any algorithm attempting gradient descent (FA given as an example). Would other additions make this point more clear?

---

### Official Review · Reviewer_Ggrt · 2024-11-04

**Soundness:** 4
**Presentation:** 3
**Contribution:** 4
**Rating:** 8
**Confidence:** 4

**Summary:**

Starting from natural gradient descent, the authors show that correlations in data can cause non-orthonormal relationship between the model's parameters. To mitigate this, they propose a decorrelation mechanism that uses only local information. They demonstrate that this mechanism improves BP accuracy, but also notably improves results for alternate training methods such as feedback alignment, and node perturbation.

**Strengths:**

- The focus on the input correlation part of natural gradient descent is novel, as is the proposed decorrelation mechanism.
- The observation that this improves performance for BP alternatives is very significant, since these methods have the potential to be more efficient for training
- The narrative exposition was easy to follow, and the concepts are explained very clearly
- The (almost) empirical evaluation supports the conclusions of the paper
- connections to biology is interesting

**Weaknesses:**

- The discussion of methods from computational neuroscience is very cursory, and would benefit from more details such as forms of specific learning rules
- The recurrence aspect of the decorrelation rule could be discussed in more depth in the main text.
- It's not clear if the proposed decorrelation method only has a recurrent formulation, or can be efficiently implemented without recurrence as well.

**Questions:**

- What is the computational cost of the proposed decorrelation method?
-  What are the implications of having a recurrent formulation for computational efficiency and implementation?
- There are second order methods which seem to have gotten some traction such as shampoo [1, 3] and K-FAC [2]. The statement in the first paragraph against 2nd order methods needs more nuance.

[1] Vineet Gupta, Tomer Koren, and Yoram Singer. Shampoo: Preconditioned stochastic tensor optimization. In International Conference on Machine Learning, pp. 1842–1850. PMLR, 2018.
[2] Yi Ren and Donald Goldfarb. Tensor normal training for deep learning models. Advances in Neural Information Processing Systems, 34:26040–26052, 2021.
[3] Vyas, N., Morwani, D., Zhao, R., Shapira, I., Brandfonbrener, D., Janson, L., and Kakade, S. (2024). SOAP: Improving and Stabilizing Shampoo using Adam. Preprint at arXiv, https://doi.org/10.48550/arXiv.2409.11321 https://doi.org/10.48550/arXiv.2409.11321.

---

> ### Author Response · Authors · 2024-11-15
> **Review Acknowledgement and Rebuttal**
>
> First, we thank the reviewer. Your summary and strengths section make clear that you have recognised the core aspects of our contributions and this is appreciated by the authors. Below we attempt to address the weaknesses and questions in a numbered order. Note that we have also uploaded a revised copy of the manuscript.
>
> ### _Weaknesses_:
> 1. We have extended our discussion of existing computational neuroscience methods for decorrelation in Section 3.2. See revised manuscript L295-L313.
>
> 2. Indeed our discussion of the recurrence aspect (and alternative biological implementations as alluded to in point 3) are rather limited. Therefore, we have extended Figure 2 to explain how our specific algorithm for decorrelation has two precisely equivalent implementations: one which is based upon a single weight matrix multiplication and one which is implemented as the fixed-point of a system of dynamics. The text has also been updated such that, in Section 3 we state
> _“Here we present a method which bridges across all of the above, allowing fast and stable decorrelation while having two equivalent formulations: one via a single weight matrix multiplication, and another as the fixed-point of a recurrent system of dynamics.”_
> Furthermore, we add greater detail to section 3 and extend the final paragraph of section 3.3 to express the utility of a decorrelation mechanism which can be expressed in a system of dynamics.
>
> 3. We hope that our additions to the text and figures in the previous point clear up any remaining questions in this context, but are happy to hear from the reviewer if not.
>
> ### _Questions_:
> 1. Indeed the computational complexity of our method was not discussed at length in the text, we have added an additional two paragraphs to the end of Section 3.2 (L388-L402) in order to explicate that our linear transformation-based decorrelation adds a number of parameters to each layer’s computation which are quadratic in the number of nodes at that layer. Therefore, such methods are particularly suitable for thin and deep rather than wide and shallow networks.
> 2. The impact of a recurrent implementation is highly dependent upon the system within which the recurrence is simulated. To be clear, we point out that we are aiming for the fixed point of our recurrent system for a decorrelated state and we describe what the key factors of consideration are within the text, again at the end of Section 3.2.
> 3. Thank you for this contribution - we have included your citations to the main introduction as a counterpoint to historical observations on second-order optimization and point out that ‘recently developed second order methods show significant promise \citep{Gupta2018-co, Ren2021-pn, Vyas2024-bh}’.
>
> We hope that these answers satisfy your questions, and we are happy to hear any additional feedback.

---

### Meta-Review · Area_Chair_ofpz · 2024-12-20

**Metareview:**

This paper investigates how correlations in data and internal activations affect the geometry of parameter updates in neural networks, extending the idea of natural gradients and linking it to the need for decorrelating transformations at every layer. While the narrative is well-structured, the work falls short on both novelty and empirical validation. Overall, the paper is a well-written synthesis rather than a substantive advancement in the field, and lacks the strong justification, new methods, or comprehensive empirical evidence needed for acceptance.

Given the variance in the reviews, the AC read the paper in entirety before making a decision.

**Additional Comments On Reviewer Discussion:**

The AC mostly agrees with the objections raised by reviewer qqzv. In particular, the response by the authors to a specific question of qqzv is: "Perhaps the missing link in this case is that we intend to give a clear theoretical explanation of why decorrelation helps in training convergence speed for BP and why it helps so much for bio-plausible algorithms. Therefore, our explanation of natural gradients is actually presented to give an insight in this regard: that decorrelation is helpful because it removes half of the difference between natural gradient descent and regular gradient descent." The AC found this explanation, and the corresponding discussion in Section 3.3 insufficient and unclear.

---

### Decision · Program_Chairs · 2025-01-22

Reject